# Is the Qi experience related to the flow experience? Practicing qigong in urban green spaces

**Shih-Han Hung[1], Ching-Yung Hwang[2], Chun-Yen Chang** [1]*

1 Department of Horticulture and Landscape Architecture, National Taiwan University, Taipei, Taiwan,
2 Department of Athletics, National Taiwan University, Taipei, Taiwan

* cycmail@ntu.edu.tw

## Abstract

People experience a healthy energy gained from the environment and an inner feeling, called the Qi experience. The flow experience has been a popular topic in Western studies, especially within the fields of psychology and health, and in all kinds of activities. Our current study used quantitative and qualitative methods to analyze the relationship between the Qi experience and the flow experience. After collecting data using open-ended questions, we integrated and connected the Qi experience into five orientations: (1) the feeling of Qi; (2) the mind; (3) Qi and consciousness; (4) physical, mental, and spiritual benefits; and (5) the feeling of Tao. The results revealed a high level of consistency between the flow experience and the Qi experience ($r = 0.90$, $p<0.00$, which supports the conclusion that the concept of the flow experience in Qigong activity seems to be the same as that in East Asian disciplines, called the Qi experience.

## Introduction

The Qi experience is the inner feeling that allows Qigong practitioners to adjust their breathing and consciousness, exercise the mind–body, and use consciousness to guide the flow of vital body energy (Qi energy) through the energy pathways (meridians) to maintain the essential health of the body-mind-spirit, which is related to achieving a deep and optimal state [1–3]. Western sciences consider Qigong to be a combination of meditation, breathing, and physical movements that is used to manage the vital energy (Qi) in the body and improve spiritual, physical, and mental health [4,29]. Enz mentioned that Qigong masters feel a *sense of Tao—* the laws of balance that harmonize the rhythms in humans and nature [2,5]—when they lose self-consciousness while practicing Qigong; in the same way, concerning Eastern Philosophy, Chuang Tzu stated that the proper way to live without concern for external rewards is enjoyment in untroubled ease, which is the essential connection between happiness, enjoyment, and even virtue [6]. In Taoist philosophy, controlling self-consciousness is a way to "entice the Qi to flow", which has similar concept of central controlling one's mind and unity with one in flow experience [7]. A flow experience is the state of focusing on an activity, which leads to an optimal experience [8]. When one balances high levels of both skill and challenge, he/she may

**Funding:** Funding was received by Dr. Chun-Yen Chang to support this study from National Taiwan University of Cutting-Edge Steering Research Project [Grant Number 10R70613, http://research. ord.ntu.edu.tw/en/DivisionStrategic.aspx]. The funder play no role in study design, data collection and analysis, decision to publish, or preparation of the manuscript.

**Competing interests:** The authors have declared that no competing interests exist.

feel action–awareness merging, loss of consciousness, and even a sense of being immersed in the environment. We generally consider the state of the flow experience in psychological attributions as the state in which one concentrates on one thing, with the mind automatically doing the movement, feeling the current consciousness, merging with the current state and environment, and then receiving immediate feedback. These psychological states might be similar to what occurs when practicing Qigong.

On the other hand, Qigong, Tai Chi, and yoga are ancient mind–body practices that all share the same three regulators in self-consciousness: body focus (posture and movement), breath, and mind focus (meditation), which benefit physical and psychological health and well-being [1,2,9,10]. Qigong focuses on balancing "Qi" in our body and mind and relaxing to achieve an optimal state of oneness [2]; however, Tai Chi refers to the martial art of meditative movement exercise [11]. The deep breath form of focusing attention on the present moment is similar to any kind of meditation, such as Qigong, Tai Chi, Yoga, or Zen activity, which all share a static form of deep meditation that evokes optimal experience in practicing [3,12–14].

In the scope of the present study, we attempted to understand the subjective experience of the relationship between the Qi experience and the flow experience, which are both part of the mental state that influences mental health, well-being, and self-consciousness. To illustrate this point, we use Qigong sensory experience to discuss the overall Qi and flow experiences. We do not focus on discussing the definition of Qi, which has roughly the same scope as Feng Shui, influential in traditional Chinese landscape design, but we do focus on the overall psychological state in the experience of Qi. The psychological mindset of the Qi experience produced in Qigong is similar to the flow experience; therefore, it is the subject of our interest.

## Qigong and the essence of the Qi experience

Qigong is an ancient type of mind–body exercise that emphasizes a harmonious interaction between humans and the Qi energy in the environment through breathing or movement, but it also plays a special role in traditional Chinese culture and medical healing and health exercise. In practicing the experience of Qi, natural surroundings, such as the movement of fresh air, wind, and tress, play important roles to connect body, self, emotion, and environment, especially in the process of "breathing" [1,15]. According to the Taoist view, the dynamic Qi energy, Yin and Yang, is a natural phenomenon in our body and in the environment. As one balances Qi, Yin, and Yang, he/she may become aware of Tao—the law of nature [5].

As Qigong practitioners calm down, they relax their body and mind and comply with their surroundings so they can experience Qi. Qi is not part of the imagination [1]—that is, practitioners feel a warm or tingling sensation of Qi energy floating in their blood, called the Qigong state [1,2,5]. From the empirical evidence-based description of the Qigong state, Qi involves the raising of the palm temperature, which is detected by far-infrared radiation [9], and changes in brain activity in the Qigong state, detected through fMRI [16], increasing the alpha and frontal theta in the Qi state [17]. On the other hand, the optimal experience might be found in the Qi experience. A systematic review of psychological well-being outcomes showed that Qigong reduces depression and anxiety and improves mood, self-efficacy, and quality of life [18]. However, what is the Qi (psychological) experience in the present? A quantitative study [19] described the inner psychological aspects of Qi as "the experience of Qi energy," "Emotion." "Attention, Concentration," "Body-Mind awareness," "Health benefits," and "other benefits." These descriptions of the Qi experience represent the main core of our study.

## The concept of the flow experience

The flow experience was defined by Csikszentmihalyi, who indicated that whatever activities people involve themselves in, when the activities go smoothly, they experience similar feelings [8]. This is referred to as the flow experience. It comprises people and the environment in a dynamic system, and interactions between people and the environment or a task. The flow experience is a state of task-absorption, cognitive efficiency, and intrinsic enjoyment that makes one completely involved in an activity, balancing the challenges of the activity and the skills of the individual [8,20]. The nine components of the flow experience were recently classified into two categories: condition (Clear Goals, Challenge–Skill Balance, and Unambiguous Feedback) and the subjective state of being in the flow (Autotelic Experience, Concentration on the Task at Hand, Paradox of Control, Action–Awareness Merging, Transformation of Time, Loss of Self-Consciousness) [21]. These categories both influence Qigong practice to achieve a sense of Qi and flow states.

On the other hand, studies have discussed using electromyography to detect one's emotional state and cortisol levels to measure cardiovascular stress, while fMRI has been used to connect flow and related optimal brain activations with a wide variety of activities [22]. Practicing meditation could be linked to the flow experience, but very few studies have addressed this connection. Hatha yoga is the oldest Eastern method for training and controlling one's mind and body and for inducing happiness. The inner feeling in Hatha yoga is similar to the flow experience [23]. Studies on yoga practitioners have reported a higher flow experience than in other physical activities chosen by participants [24]. Yoga, flow experience, and mindfulness can promote a positive mental state and decrease anxiety due to total immersion in the activity [13].

## A comparison of the flow and Qi experience concepts

There has thus far been relatively little research on the Qi and flow experiences. From the perspective of meditation, we argue that Qigong activity could evoke the psychological flow and Qi experiences. In Eastern disciplines, martial arts influenced by Taoism and Zen Buddhism, such as Tai Chi, Kung Fu, and Judo, represent special forms to evoke the flow experience [23]. These activities all emphasize consciousness-controlling skills to improve the mental and spiritual state that are reminiscent of the concept of the flow experience. From the emotional perspective of practicing Qigong as related to the flow experience in four weeks of training, the results of content analysis described the affective state through open-ended answers as restful, positive affect, balanced, and lucid; the results of self-report questionnaires using the Positive and Negative Affect Scale (PANAS) showed that Qigong evoked a positive affective state, with participants being in the flow state for 20 minutes, then 40 minutes, then 60 minutes [25].

As one meets the condition of the flow, he/she may subjectively be in the flow. The subjective state in the flow includes the merging of action–awareness, the transformation of time, the loss of self-consciousness, and autotelic experience, a state which could be described as similar to the Qi experience. Up to this point, we have provided our explanation to assess the possible contribution of comparing the Qi and flow experiences via qualitative studies [19] (see Table 1). Table 1 shows a systematic comparison system for evaluating construct validity in the Qi and flow experiences.

## Aim and scope of the study

Qigong, Tai Chi, yoga, and other meditation activities decrease depression and anxiety and increase positive emotions and well-being. Researchers have focused primarily on complementary and alternative medicine or on using randomized controlled trials to discuss positive

**Table 1. A systematic evaluation of the construct validity of the Qi and flow experiences.**

| | Qi experience [1–3,19] | Flow experience [8,20,26] | Concept comparison |
|---|---|---|---|
| **Definition** | A Qi experience is the inner feeling that allows Qigong practitioners to adjust their breathing and consciousness to guide the flow of vital body energy (Qi energy) through the energy pathways and maintain the essential health of the body-mind-spirit. | The flow experience is a state of task-absorption, cognitive efficiency, and intrinsic enjoyment that makes one completely involved in an activity, balancing the challenges of the activity and the skills of the individual. | As one is totally absorbed in the activities, balancing skills and challenges that lead to feeling a sense of Qi, one may gain optimal experience. |
| **Traits** | **The experience of Qi energy**<br>• energy flows easily<br>• feeling energy as the participant is trained<br>• perceiving energy moving in the body as a state, not in the imagination | **Autotelic experience**<br>• self-affirmation, self-growth, pleasure, satisfaction, quality of life, well-being, and more as positive psychological benefits<br>**Challenge–skill balance**<br>• When one balances skills and activity challenges, one is more likely to achieve a sense of flow<br>**Paradox of control**<br>• When the activity goes smoothly, a sense of control can be generated, one which can control the activity and even the environment of the activity at the moment; this sense of control does not involve consciousness, but rather a feeling that one can handle everything | As one completely focuses on balancing "Qi" in the body and mind, the Qi energy is experienced. It is similar to the concept of *challenge–skill balance*, since as one balances the skills and challenges of the activities, a sense of flow may be achieved. Furthermore, the descriptions of the Qi experience are similar to the concepts of *autotelic experience* and *paradox of control* in the flow. |
| | **Emotion**<br>• positive emotion, happiness, enjoyment, satisfaction after performing well<br>• decreased negative emotions<br>• less stress<br>• happiness<br>• internal harmony<br>• somewhat rejuvenated<br>• pure essence of the soul, something deeper | **Autotelic experience**<br>**Unambiguous feedback**<br>• After setting up<br>• clear goals, one can successfully complete the goals and clearly feel the feedback from the activity<br>**Loss of self-consciousness**<br>• When the activity goes smoothly, the participant enters the world of "the self" and does not care about others | The *autotelic experience* and *unambiguous feedback* in the flow experience are that "in the state of practicing" and "after practicing," respectively, people can gain immediate rewards, such as a good mood, enjoyment, and satisfaction, observable in both the flow and Qi experiences.<br>The *loss of self-consciousness* is about losing oneself and merging with one's surroundings without concern about presentation. However, the Qi experience is a state of relaxed awareness with breathing that evokes a sense of deep spiritual feelings, interactions with nature, and even experiencing the Tao. Therefore, the optimal state seems to be the same as that described in the concepts of flow and Qi in this study. |
| | **Attention, Concentration**<br>• concentrate on energy, experiences, breath and movements<br>• attention focuses on movements, movement follows thoughts<br>• sense of "losing contact" with reality, forgetting about life's troubles and worries | **Loss of Self-consciousness**<br>**Concentration on the task at hand**<br>• When one concentrates on the activities, one can ignore other things without distractions<br>**Clear goals**<br>• Have a strong sense of knowing what to do, focusing on the goal, and forgetting annoying chores | As one concentrates on one thing, knows the goals, and controls one's thoughts and actions without worries, the movements merge with the body, and one can feel Qi energy floating in the body and a sense of being lost. These concepts are highly similar to *loss of self-consciousness*, *concentration on the task at hand*, *clear goals*, and *action–awareness merging* in the flow experience. |
| | **Body–Mind awareness**<br>• better quality, fluency, and smoothness of movement<br>• deeper consciousness<br>• feel the body's position in space precisely in every moment<br>• movements automatic, meditation-like state | **Loss of Self-consciousness**<br>**Action–awareness merging**<br>The deeper one is involved in the activity without worries, the higher the coordination and likelihood of spontaneously completing the action | |
| | **Health benefits**<br>• obtain health-related benefits<br>• feel energy and reduce health imbalances in the body<br>• feel better, even much better than normal<br>• experience vitality, wellness, and life energy<br>• decreased stress or tension<br>• relax each muscle with the movement, promoting a sense of emotional wellness and, for example, happiness, joy, or other improved mood | **Autotelic experience**<br>**Unambiguous feedback** | Qi participants feel the Qi state floating in the body and gain physical, mental, and spiritual refreshment and recovery. This *unambiguous feedback* and *autotelic experience* from practicing could lead to a Qi experience and to a flow experience. |
| | **Others**<br>• Memory<br>  • memorize movement patterns,<br>  • sequences of movements seem to be challenged<br>  • be distracted, as one cannot remember the movement; be a benefit, as one can remember the movement<br>  • internal order helps memorize the movements<br>• Peak performance<br>  • minimum effort and maximum concentration<br>  • remember years after practicing the perfect performance intensively or nervously thinking about the next movement; does not mention peak performance | **Autotelic experience**<br>**Concentration on the task at hand**<br>**Challenge–Skill balance** | *Concentration on the task at hand to achieve challenge–skill balance* leading to *autotelic experience* could indicate that as one goes into the Qigong state, he/she can feel a sense of peak performance and attain fluent breathing and movements. |
| | | **Transformation of time** | The concept of participating in an activity that seems to encourage the feeling of a "different time flow than usual," which could happen in a deep flow. |

physical and psychological outcomes, such as positive cardiopulmonary effects, improved psychological symptoms, improved immune function and reduced inflammation, beneficial mood changes, and a pleasant emotional state, as opposed to discussing the similarity between psychological concepts in the Qi experience and the flow experience. The psychological phenomena in the mind–body of the Qi experience influence attention, enhance positive emotion, and increase mind–body awareness via the perception of Qi energy moving in the body, especially when interacting in a green environment. This study concerned the psychological states that might be similar to what occurs when totally focusing on practicing Qigong to achieve the Qi experience and the flow experience while practicing in an urban green space. In light of these concerns, the purpose of this article is to describe the relationship between the Eastern concept of Qigong, which induces the Qi experience, and the Western concept of the flow experience as evoked by Qigong. We used a qualitative methodology to generate the Qi experience questionnaire and a quantitative methodology to test the correlation between the Qi experience and the flow experience. More specifically, this study was undertaken to determine whether the Qi and flow experiences refer to the same concept in several respects during the practicing of Qigong in an urban green space.

## Methods

### Developing the Qi experience questionnaire (QEQ)

The Qi experience in this study was defined as the state of perceiving the Qi experience in the body-mind-spirit while practicing Qigong. We invited students who have practiced primary Qigong as a sample to survey their inner feelings. The study asked the sampled students to, in one sentence or adjective, describe their Qi experience by answering the open-ended questions. Data were collected on 12 occasions during the late spring and late fall (at the end of the semester) from students we considered to be more familiar with the Qi experience. Each primary Qigong course included 26 students who, after practicing, were asked to respond to the open-ended questions on the questionnaire twice in three different practice environments: (a) indoors, (b) the built environment within natural surroundings, and (c) the natural environment, beside an ecological pond (see Figs 1–3). Based on Table 1, participants described their Qi experiences as "feeling energy in the body," "concentrating on breath and action," "losing contact with reality," "decreased stress," and "a peak experience".

The Qi experience questionnaires were actively discussed with three Qigong masters, who had practiced for over 10 years, and with three researchers to rephrase the descriptive sentences or adjectives into one sentence. We identified the similar concepts and integrated and connected them into five categories (and a total of 14 items) (see in Table 2): (1) the feeling of Qi (Cronbach's alpha = 0.90), (2) the mind (Cronbach's alpha = 0.93), (3) the Qi and consciousness (Cronbach's alpha = 0.86), (4) the physical, mental, and spiritual benefits (Cronbach's alpha = 0.89), and (5) the feeling of Tao (Cronbach's alpha = 0.91) [27,28].

In order to justify the Qi traits according to Tables 1 and 2, we used the multi-trait-multi-method (MTMM) matrix [29,30] to assess the convergent validity of the question, "what is Qi," and the discriminant validity of Qi traits (Table 3). The 1–3 questions in the *feeling of Qi* category tested one's ability to sense the Qi floating in the body, smooth and unhindered, which signifies entering the state of Qi. The convergent validity in the correlations matrix was $r = 0.70$ to $r = 0.81$ and was slightly different from the other traits ($r = 0.60$–$0.77$). The 4–7 questions in the *mind* category refer to a feeling of tranquility, happiness, and relaxation during the Qi experience. The convergent validity in the correlations matrix was $r = 0.71$ to $r = 0.81$ and was slightly different from the other traits ($r = 0.62$–$0.79$). The 8–9 questions in

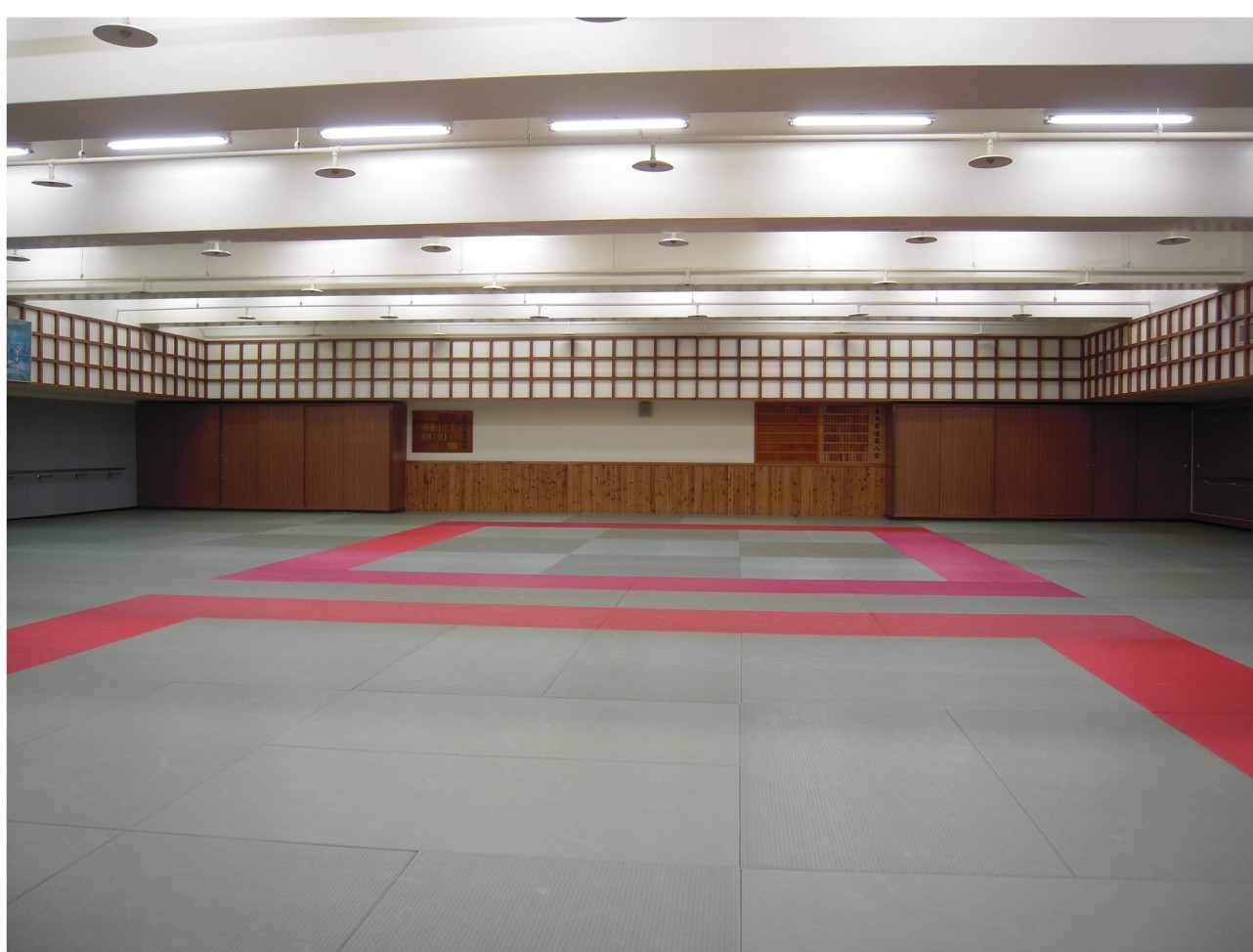

**Fig 1. The indoor location where primary students usually practice Qigong.**

the *Qi and consciousness* category focused on the state of consciousness and concentration and had a correlation coefficient of 0.76, but it was slightly difficult to distinguish between the traits ($r = 0.67–0.83$). The 10–12 questions in the *physical, mental, and spiritual benefits* category focused on recovery of the mind and spirit, yielding an acceptable correlation coefficient ($r = 0.69–r = 0.78$) that was slightly different from other traits ($r = 0.65–r = 0.76$). The 13–14 questions in the *feeling of Tao* category were taken from a concept in ancient Eastern disciplines that refers to one who can understand the harmony between man and nature, for which the correlation coefficient of convergent validity was high ($r = 0.83$). Table 3 shows high reliability and convergent validity for each of the Qi experience traits, however, slightly distinguish between other traits. In the Taoist view, Qigong is an activity undertaken to balance mind-body-spirit and the dynamic of Ying/Yang states, and the movement of Qi floating in the body is related to holistic experience [4,5,31].

The total internal reliability of Cronbach's alpha on the Qi experience questionnaires was 0.97. A five-point Likert scale was used to measure the Qi experience, with a higher score meaning a greater experience of Qi.

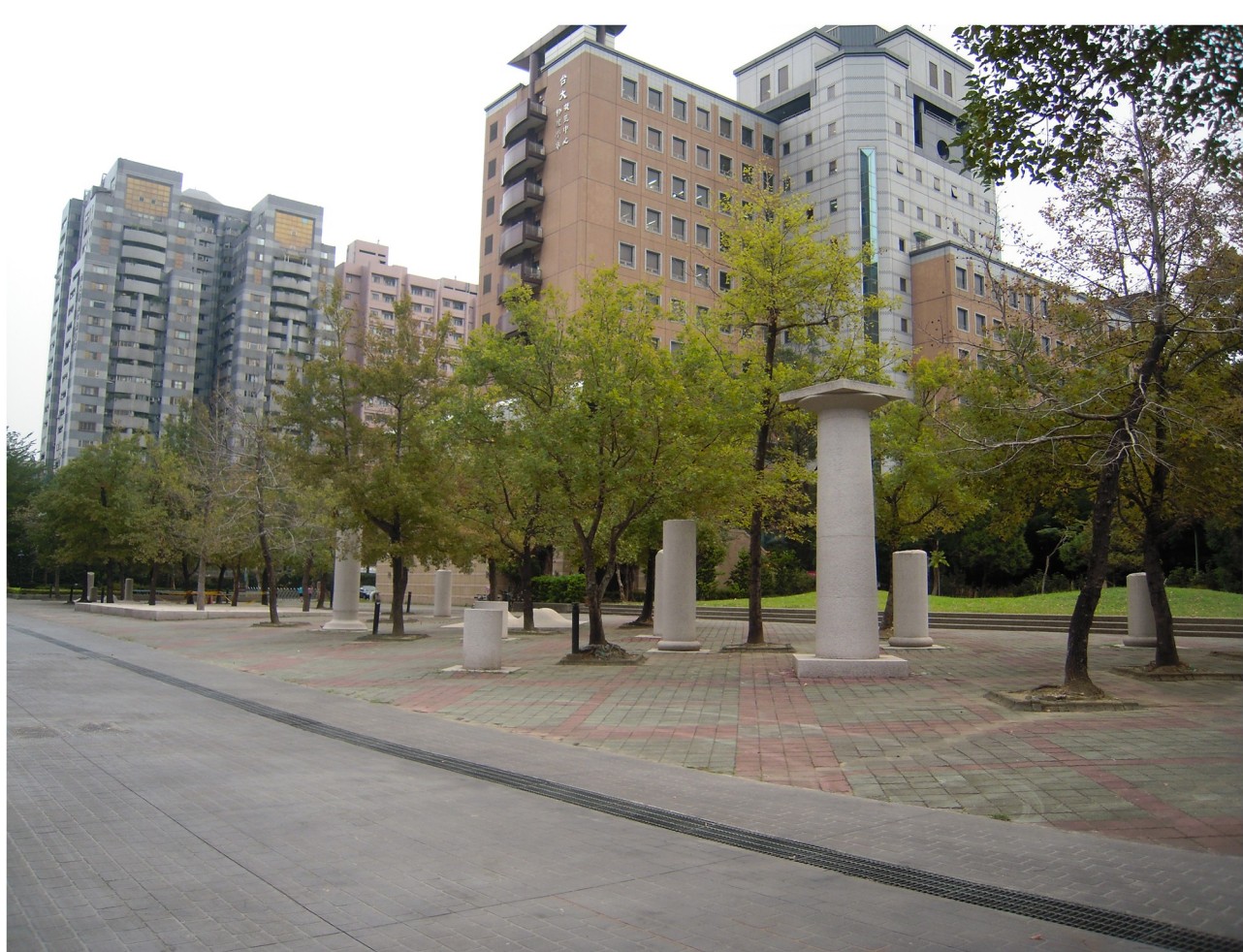

**Fig 2. The built environment within natural surroundings where primary students usually practice Qigong.**

## Instruments of the Flow State Scale (FSS)

In this study, the flow experience was measured as the challenge–skill balance that one achieves to gain an optimal experience state, which can be evoked by practicing Qigong. Jackson and Marsh developed the FSS [26], which showed acceptable levels of reliability (Cronbach's alpha = 0.93), consisting of 36 items which were based on the nine dimensions of flow: challenge–skill balance, merging of action–awareness, clear goals, unambiguous feedback, concentration on the task at hand, sense of paradox of control, loss of self-consciousness, transformation of time, and autotelic experience [8]. *Action–awareness merging* refers to thoughts automatically following the movements as practiced. *Unambiguous feedback* emphasizes that by setting clear goals and completing them, people may get immediate feedback from the activities. *Transformation of time* means that participants may feel the passage of time differently. *Autotelic experience* refers to the concept of achieving self-affirmation, self-growth, pleasure, satisfaction, quality of life, well-being, as positive psychological benefits. Research has shown that most practitioners feel *autotelic experience*, *concentration on the task at hand*, *sense of paradox of control*, *challenge–skill balance*, and *unambiguous feedback*; however, only the few people who feel *transformation of time* and *loss of self-consciousness* can experience the state of deep flow [32]. Abbreviated descriptions of the items on the FSS were used to measure

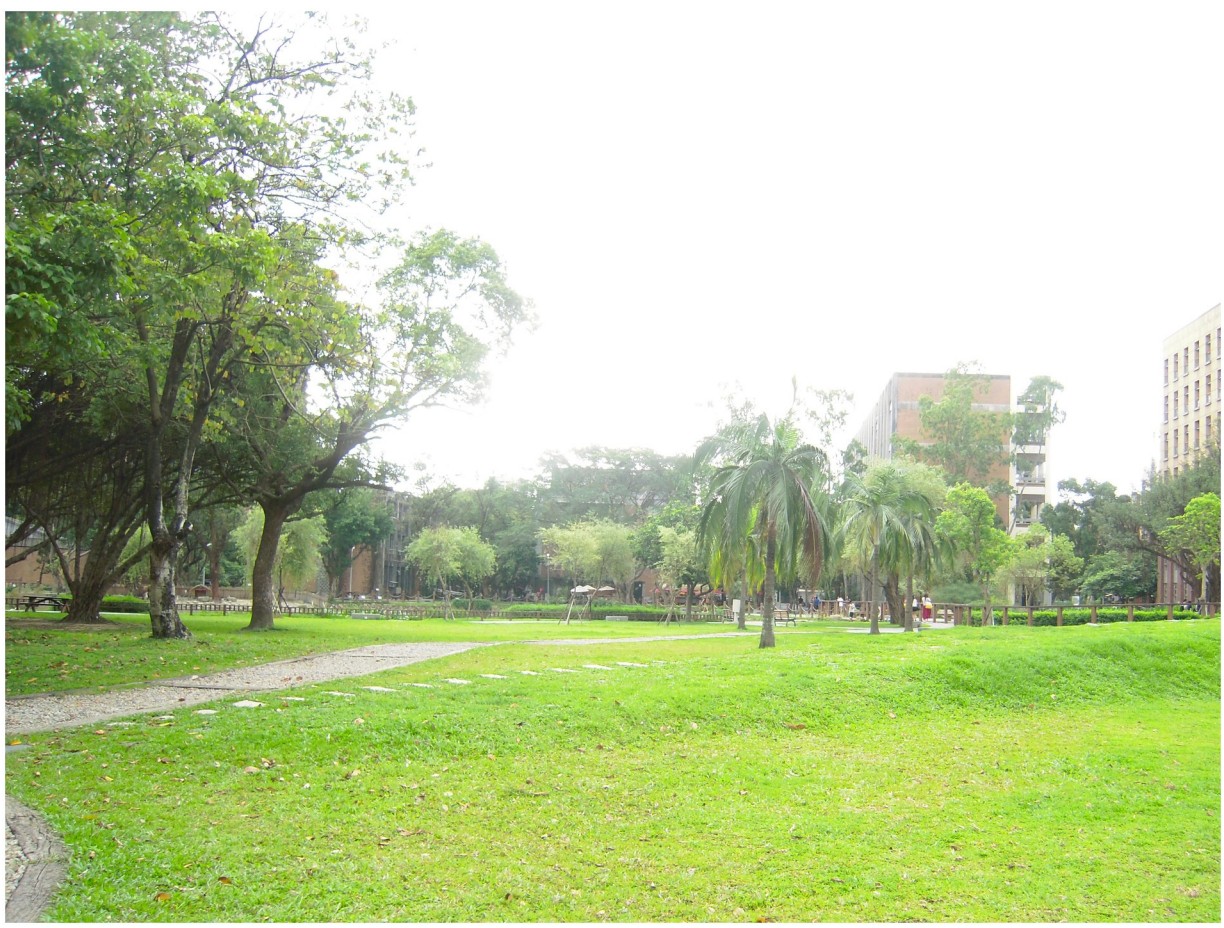

**Fig 3. The natural environment, beside an ecological pond, where primary students usually practice Qigong.**

the flow experience in sports players [32]. These items were then translated into Chinese, with the items "Qi experience," "Qigong practice," or "Qi" added to describe the active state of the flow experience, which can be induced by practicing Qigong. The modified sentences, however, retained the main statement of the concept. For instance, "experience is extremely rewarding" is the main concept in *autotelic experience*; therefore, the question was modified as "I think practicing Qigong is extremely rewarding in this time." Moreover, the original abbreviated description for the concept of *loss of self-consciousness* included *not concerned with others* and *not worried about others*. However, in the current study, due to the extended definitions of *concerned* and *worried* and attempts to understand the interaction between the environment and humans, the question was revised from *not worried about others* to *feeling of integration with the environment*. The revised version of loss of self-consciousness showed acceptable reliability (Cronbach's alpha = 0.79).

For the content validity of the revised FSS, we invited three Qigong novice practitioners to review the comprehensibility of each item, after which we invited three Qigong masters to determine the knowledge and skill level of the content tested in each question. We invited 58 subjects to test the revised FSS, which showed very good reliability (Cronbach's alpha = 0.98) and used a five-point Likert scale, as seen in Table 4. The higher the score a participant achieved, the deeper his or her flow experience gained from Qigong.

**Table 2. A Fourteen questions used to measure the Qi experience based on the open-ended questionnaire responses.**

| Qualitative method | Quantitative method |
|---|---|
| Open-ended data about the Qi experience | 14 questions used to measure the Qi experience (Cronbach's alpha = 0.97) |
| **Describing the Qi experience:**<br>• Feel the subtlety of the Qi experience<br>• Hard to control the Qi experience<br>• The obvious presence of Qi energy floating in the body<br>• Strong Qi experience<br>• No sense of Qi; Less sense of Qi energy; No activation of Qi; No special feeling; Weak Qi; feel good but don't know how to master Qi<br>• Feel the Qi floating<br>• Feel a good sense of Qi<br>• Sense of Qi | **Feeling of Qi (FQ)**<br>1. In this practice, I feel a good sense of Qi. |
| • Smoothly practicing Qi<br>• Qi is concentrated in one spot but cannot easy flow into other spots<br>• Feel more concentrated on Qi floating<br>• A little magical; weird<br>• Difficult to control the Qi floating | 2. In this practice, I feel I am smoothly practicing Qi. |
| • Spontaneous and automatic production of Qi feels scary but relaxing<br>• The traditional environment of Qi is distributed in this practice, maybe the predecessors' Qi exists here<br>• Easy to go into the Qi state<br>• I feel that the Qi energy is much stronger in the natural environment<br>• Feel a stronger Qi energy in the outdoor environment and makes the exercises more fluent<br>• Strong Qi feeling in an outdoor environment | 3. In this practice, I feel my body and mind may quickly go into the state of Qi. |
| **Mind:**<br>• Sometimes calm, sometimes messy<br>• Quiet and peaceful<br>• Calm<br>• Easy and stress-free<br>• Smooth flow | **Mind (M)**<br>4. In this practice, I feel tranquility. |
| • Comfortable<br>• Relaxed<br>• Leisurely<br>• Happy<br>• The overall feeling is quite comfortable, but the degree of practice needs to be strengthened<br>• No feeling anxious, uneasy, doubtful, stressful<br>• Confused, trance-like<br>• Good mood, pleasant<br>• Enjoying the moment | 5. In this practice, I feel happy and comfortable. |
| • Release pressure<br>• Release stress<br>• No anxiety | 6. After practicing, I think my pressure may be released. |
| • Restructure thoughts | 7. After practicing, I think I can restructure my thoughts. |
| **Qi and Consciousness:**<br>• Feeling that Qi energy is more concentrated<br>• Mindlessness, no distractions<br>• Focus on the state of Qi<br>• Cannot concentrate on practice<br>• Quiet and with no worries | **Qi and Consciousness (QC)**<br>8. In this practice, I can focus on my consciousness. |

(*Continued*)

**Table 2.** (Continued)

| Qualitative method | Quantitative method |
|---|---|
| • Thoughts follow actions<br>• Clear goal to make thoughts follow the Qi | 9. In this practice, I feel my thoughts follow my actions. |
| **Physical, mental, & spiritual benefits**:<br>• Full of energy<br>• Good mental state<br>• Spiritually relaxed<br>• The same practice as usual, but feels much better in mental recovery<br>• The progress is not ideal, but with more practice, the body and mind will relax<br>• Physical and mental relaxation | **Physical, Mental, and Spiritual Benefits (PMSB)**<br>10. After practicing, I feel full of energy in body-mind-spirit. |
| • Recovery of body and mind<br>• Relaxed and happy, refreshing body and mind | 11. After practicing, I can totally refresh my body and mind. |
| • Whole body is relaxed and comfortable<br>• Sense of being healthy<br>• Body relaxed | 12. After practicing, my whole body relaxes and is more comfortable. |
| **Tao**:<br>• When practicing, I seem to be too eager to concentrate on the spirit<br>• Follow a clear heart<br>• Govern by doing nothing that goes against nature (無為而治、順其自然)<br>• Be happy and pleased with nature | **Feeling of Tao (FT)**<br>13. In this practice, my spirit is clear and has a deep understanding of nature. |
| • Unity of heaven and human; Tao follows nature<br>• Realize the Tao | 14. In this practice, I can feel the meaning of "Tao follows nature" (道法自然). |
| **Others**<br>**1. Personal status**:<br>• Not feel comfortable today<br>• Dizzy, uncomfortable, headache<br>• Cannot concentrate on practicing due to personal physical condition<br>• Lack of sleeping, feeling less active today<br>• Went to bed too late, cannot practice well<br>• Do not have enough sleep, feel dizzy, and use Tai Chi movements to reduce pain<br>• Too tired<br>• Anxious because of the exam<br>• A further understanding of the body<br>• Not meeting goals | A few subjects mentioned problems with their personal routine, such as a lack of sleep, that might influence Qi practicing; however, these statements are not integral to describing "the state of perceived Qi experience in body-mind-spirit." |
| **2. State of practicing skills**<br>• No practice and forgetting the skills<br>• Not familiar with the skills, but feel good<br>• Skills have improved<br>• It's better to practice<br>• More familiar with the skills<br>• There is room for improvement<br>• Skills may go even further, but there is not yet much of a breakthrough<br>• Not satisfied with the skill, but it is worth it to practice and make the body and mind more relaxed. | The state of practicing skills is considered as one's personal skills meeting challenges, but the statement would be too similar to the flow experience (Q9–12 in FSS [Table 3]) for participants to discriminate. Therefore, we did not use this in the questionnaires. |

(*Continued*)

**Table 2.** (Continued)

| Qualitative method | Quantitative method |
|---|---|
| **3. Environment**:<br>• Hot outside, much sunshine<br>• Can feel the sunlight<br>• The environment is bright and relaxed without burden<br>• Outdoors environment is better than the indoors<br>• Sunny and warm<br>• Can feel stronger Qi in outdoor environment<br>• Feel stronger Qi in outdoor environment<br>• It's windy and a little bit cold in outdoor environment<br>• Indoor classroom is where we usually practice and feel a sense of safety | The research time we chose was late fall in the indoor environment (Fig 1) and in the outdoor environment (Fig 2) about 25.7˚C and 20.8˚C; outdoor environment (Fig 3) is about 23.8˚C and 26.1˚C; in late spring outdoor environment (Fig 2) is about 30.8˚C and 33.3˚C; outdoor environment (Fig 3) is about 29.4˚C and 25.1˚C. The data were measured by the Central Weather Bureau.<br>When asked for one sentence or adjective to describe their Qi experience, subjects mentioned that by practicing in outdoor environments, especially within nature, they can feel a stronger Qi experience than indoors. However, a few subjects thought the outdoors was a little too hot or cold. |
| **4. Time**:<br>• Time passes quickly; time is slow<br>• Time passes slowly and differently than usual<br>• The Qi practicing time is too short but comfortable<br>• The Qi practicing time is too short to stay in the Qi state<br>• The Qi practicing time is too short to feel the Qi | Time passing slowly and differently than usual is viewed as "transformation of time" in the flow. However, the statement of transforming into questionnaire would be too difficult for subjects to discriminate the psychological response in the Qi or flow questionnaires. Therefore, we chose to use the flow questionnaires instead |
| 5. Others:<br>• The movement generated is natural and comfortable without thinking | This statement, concerning generating natural movement without thinking, is similar to the "paradox of control" in the flow experience. However, this statement would be too difficult for subjects to discriminate the psychological responses in the Qi or flow questionnaires. Therefore, we chose to use the flow questionnaires instead. |

[a]The Qi experience Questionnaire (QEQ) scale was created using statements about the Qi experience, and the results were similar to those in the qualitative study by Posadzki [19] (Table 1). We then integrated similar concepts about the Qi experience into 14 questions.

## Study site

Data were collected at the National Taiwan University (NTU), a representative of the urban green environment in Taipei City, which provides various landscape types for nearby residential exercise, health, and well-being. Besides, many Qi practitioners choose NTU campus as their practicing sites. Therefore, in this research, we selected 30 study sites, included in Figs 2 and 3, for the process of "breathing" to induce the Qi and flow experiences.

## Participants

The research complies with the main points of the research ethics statement, which is to inform the participants of the research purpose, methods and process, the benefits, the privacy of academic use and anonymous analysis, etc. for ethical considerations. In addition, we ensure the participants' adult age (20 years) in Taiwan, and then ask them to sign an informed consent form. The research ethics committee (REC) approval was not expected to apply at that time during the funding from Ministry of Education.

In this study, participants were selected based on the criterion that they could sense the Qi state, even if only weakly. We assumed our participants could feel sense of Qi state stability in order to test the relationship between the Qi experience and the flow experience in multiple Qigong practicing. Sixty-two participants were recruited from a primary Qigong course at NTU and from the Taipei Tan Tao Culture Research Association. Thirteen of the 52 students who participated in the study on perceptions of the Qi experience were willing to participate in this study. We randomly assigned the 62 participants to 12 urban green space experimental sites at NTU. However, since four participants did not complete the experiment, a total 58 of participants (NTU students = 40, Taipei Tan Tao Culture Research Association members = 18, $M$ = 33.7 years old) were measured in the statistical analysis.

**Table 3. Using the multi-trait-multi-method (MTMM) to determine the construct validity of the Qi experience.**

| | Qi1 | Qi2 | Qi3 | FQ | Qi4 | Qi5 | Qi6 | Qi7 | M | Qi8 | Qi9 | QC | Qi10 | Qi11 | Qi12 | PMB | Qi13 | Qi14 | FT |
|---|---|---|---|---|---|---|---|---|---|---|---|---|---|---|---|---|---|---|---|
| Qi1 | – | | | | | | | | | | | | | | | | | | |
| Qi2 | .81[b] | – | | | | | | | | | | | | | | | | | |
| Qi3 | .70[b] | .72[b] | – | | | | | | | | | | | | | | | | |
| FQ | .92 | .92 | .89 | (0.90)[a] | | | | | | | | | | | | | | | |
| Qi4 | .67[C] | .70[C] | .76[C] | .78 | – | | | | | | | | | | | | | | |
| Qi5 | .73[C] | .71[C] | .74[C] | .80 | .80[b] | – | | | | | | | | | | | | | |
| Qi6 | .67[C] | .65[C] | .73[C] | .75 | .71[b] | .75[b] | – | | | | | | | | | | | | |
| Qi7 | .67[C] | .67[C] | .71[C] | .75 | .72[b] | .77[b] | .81[b] | – | | | | | | | | | | | |
| M | .76 | .76 | .81 | .85 | .89 | .92 | .90 | .91 | (0.93)[a] | | | | | | | | | | |
| Qi8 | .75[C] | .77[C] | .70[C] | .82 | .69[C] | .70[C] | .71[C] | .73[C] | .78 | – | | | | | | | | | |
| Qi9 | .69[C] | .71[C] | .74[C] | .78 | .75[C] | .79[C] | .77[C] | .78[C] | .85 | .76[b] | – | | | | | | | | |
| QC | .77 | .79 | .77 | .85 | .77 | .79 | .79 | .80 | .87 | .94 | .94 | (0.86)[a] | | | | | | | |
| Qi10 | .68[C] | .68[C] | .75[C] | .78 | .72[C] | .78[C] | .78[C] | .79[C] | .85 | .74[C] | .83[C] | .84 | – | | | | | | |
| Qi11 | .70[C] | .74[C] | .67[C] | .78 | .65[C] | .68[C] | .65[C] | .67[C] | .74 | .77[C] | .71[C] | .79 | .69[b] | – | | | | | |
| Qi12 | .70[C] | .71[C] | .70[C] | .77 | .72[C] | .74[C] | .73[C] | .76[C] | .82 | .78[C] | .80[C] | .84 | .78[b] | .75[b] | – | | | | |
| PMB | .77 | .78 | .78 | .85 | .77 | .81 | .80 | .82 | .88 | .84 | .86 | .91 | .90 | .90 | .93 | (0.89)[a] | | | |
| Qi13 | .65[C] | .66[C] | .66[C] | .72 | .65[C] | .71[C] | .67[C] | .71[C] | .76 | .72[C] | .73[C] | .77 | .71[C] | .68[C] | .76[C] | .79 | – | | |
| Qi14 | .60[C] | .61[C] | .61[C] | .67 | .63[C] | .67[C] | .62[C] | .66[C] | .71 | .67[C] | .71[C] | .74 | .67[C] | .65[C] | .72[C] | .75 | .83[b] | – | |
| FT | .65 | .67 | .66 | .73 | .67 | .72 | .68 | .72 | .77 | .73 | .76 | .79 | .72 | .69 | .77 | .80 | .96 | .96 | (0.91)[a] |

$N$ = 58, $p<0.01^{**}$; Abbreviations: FQ = Feeling of Qi; M = Mind; QC = Qi and Consciousness; PMSB = Physical, Mental, and Spiritual Benefits; FT = Feel the Tao;

[a]The reliability of Cronbach's alpha is shown in the main diagonal (denoted blue);

[b]The convergent validity in each trait is shown in orange color;

[C]The discriminant validity between other traits is shown in yellow color. Significant correlations are shown in this table.

To verify participants background, willingness, and rights to join the study, they were asked to sign a Qi knowledge letter and an environmental experience informed consent form. They were then given the Qi-questionnaire package with 12 questionnaires that randomly assigned them to 12 urban green space experimental sites at NTU. The description in the consent form read as follows: *We appreciate you joining this Qi and environment experiment. This study is designed to understand your inner feelings about practicing Qigong. Our research data are only for academic use. Please pay attention to your physical condition and then practice breathing for 10 minutes at the randomly assigned experimental sites. You may practice breathing at 2–3 experimental sites per day from 7:00–11:00 in the morning, after which you may finish the questionnaire. For returning the completed Qi-questionnaire package, you will get a small gift.*

The first part of the questionnaire concerned inner feelings that described the participants' present psychological state, comprising 36 flow questions and 14 Qi questions. The second part concerned basic information, such as gender, birth year, years practicing Qigong, and date at the time of practicing. After returning the completed Qi-questionnaire package, the participants were given a 30 USD gift card as a reward.

## Data collected

Participants were asked to practice at the assigned sites from 7:00–11:00 in the morning in early spring because research [33] suggests that when the human body and internal organs contact the sunlight directly, a good Qi flow may be produced in the body. A total of 659 completed questionnaire packages were returned, for a return rate of 88.58%. Besides, all data were analyzed anonymously in this study.

**Table 4. The revised Flow State Scale (FSS).**

| Aspects | Items |
|---|---|
| Autotelic Experience (AE) (Cronbach's alpha = 0.91) | 1. I think practicing Qigong is extremely rewarding. |
| | 2. I enjoy practicing the Qigong. |
| | 3. I want to recapture the feeling of practicing Qigong. |
| | 4. The process of practicing Qigong let me feel great. |
| Clear Goals (CG) (Cronbach's alpha = 0.83) | 5. In this Qigong practice, I know what I want to do. |
| | 6. In this Qigong practice, I strongly sense what I wanted to feel from the Qi experience. |
| | 7. In this Qigong practice, I know what I need to improve. |
| | 8. In this Qigong practice, I clearly know what to do to reach my goals. |
| Challenge–Skill Balance (CSB) (Cronbach's alpha = 0.88) | 9. In this Qigong practice, I think I am competent to meet the demands. |
| | 10. In this Qigong practice, I think my abilities match the challenges. |
| | 11. In this Qigong practice, I could make the challenge and skills equally high. |
| | 12. In this Qigong practice, my skills meet the challenges. |
| Concentration on Task (CT) (Cronbach's alpha = 0.87) | 13. In this Qigong practice, I am total concentrated. |
| | 14. In this Qigong practice, I completely focused on the task. |
| | 15. In this Qigong practice, I focus my attention and put aside my worries. |
| | 16. In this Qigong practice, I keep my mind on what is happening in Qi. |
| Paradox of Control (PC) (Cronbach's alpha = 0.88) | 17. In this Qigong practice, I could completely control what I was doing in my thoughts and movement. |
| | 18. In this Qigong practice, I could easily control what I was doing. |
| | 19. In this Qigong practice, I think I could feel total control of the process. |
| | 20. In this Qigong practice, I could totally control and smoothly finish every action without thinking. |
| Unambiguous Feedback (UF) (Cronbach's alpha = 0.86) | 21. In this Qigong practice, I knew how well I was doing by the way I was performing. |
| | 22. In this Qigong practice, I clearly knew how well I obtained and controlled the state of Qi. |
| | 23. In this Qigong practice, I clearly knew how well I was doing while performing. |
| | 24. In this Qigong practice, I was aware of how well I was performing. |
| Action–Awareness Merging (AAM) (Cronbach's alpha = 0.90) | 25. In this Qigong practice, my thoughts automatically followed my actions. |
| | 26. In this Qigong practice, my thoughts and Qi were spontaneous and automatically produced. |
| | 27. In this Qigong practice, I can automatically lead my thoughts and Qi. |
| | 28. In this Qigong practice, I can do correct movements without thinking. |

(*Continued*)

**Table 4.** (Continued)

| Aspects | Items |
|---|---|
| Transformation of Time (TT)(deep flow) (Cronbach's alpha = 0.87) | 29. In this Qigong practice, I feel time differently from normal. |
| | 30. In this Qigong practice, I often feel altered time. |
| | 31. In this Qigong practice, I feel time goes slowly, and the surroundings are in slow motion. |
| | 32. In this Qigong practice, I feel time stop and do not know how long I am practicing. |
| Loss of Self-Consciousness (LSC)(deep flow) (Cronbach's alpha = 0.79) | 33. In this Qigong practice, I do not concern myself with others (I lose myself). |
| | 34. In this Qigong practice, I have a feeling of integration with the environment. |
| | 35. In this Qigong practice, I do not concern myself with presentation. |
| | 36. In this Qigong practice, I do not worry about how others look at my performance. |

## Results

### A descriptive picture of the data

A total of 659 completed questionnaires packages ($N = 58$, 11–12 times per person) were collected, including five incomplete responses. Therefore, 654 valid packages ($n = 444$ (67.9%) males; $n = 210$ (32.1%) females) were measured in the statistical analysis. A t-test of descriptive statistics presented no significant difference ($t = 0.87$, $p = 0.39$) between males ($N = 39$, $n = 444$, $M = 3.50$, $SD = 0.59$) and females ($N = 19$, $n = 210$, $M = 3.46$, $SD = 0.59$) in the flow experience; however, a significant difference ($t = 2.08$, $p = 0.04$) in the Qi experience was found based on these parameters (males [$N = 39$, $n = 444$, $M = 3.58$, $SD = 0.72$]; females [$N = 19$, $n = 210$, $M = 3.45$, $SD = 0.70$]). The ANOVA on the number of years of Qigong practice indicated no significant influence on the flow experience: $F(2, 651) = 2.59$, $p = 0.08$, $p > 0.05$, $\eta^2 = 0.01$, for those practicing under half a year ($N = 24$, $n = 276$, $M = 3.47$, $SD = 0.55$), half a year to under five years ($N = 26$, $n = 291$, $M = 3.47$, $SD = 0.60$), and six years to over 15 years ($N = 8$, $n = 87$, $M = 3.62$, $SD = 0.66$). The ANOVA also revealed a group difference concerning the Qi experience: $F(2, 651) = 5.04$, $p = 0.01$, $p < 0.05$, $\eta^2 = 0.02$), for those practicing under half a year ($N = 24$, $n = 276$, $M = 3.55$, $SD = 0.66$), half a year to under five years ($N = 26$, $n = 291$, $M = 3.47$, $SD = 0.72$), and six years to over 15 years ($N = 8$, $n = 87$, $M = 3.74$, $SD = 0.81$). The Games-Howell post hoc test showed a significant difference between half a year to under five years and six years to over 15 years. The more years the participants had been involved in practicing Qigong, the deeper the sense of the Qi experience they could feel.

### High correlation between the flow and Qi experiences

The results of the Pearson's product-moment correlation, shown in Table 5, revealed a high correlation coefficient of 0.90 at the 0.01 level ($p = 0.00$) between flow experience ($M = 3.49$, $SD = 0.59$) and Qi experience ($M = 3.54$, $SD = 0.71$). High correlation coefficients ranging from 0.77 to 0.87 ($p = 0.00$) were found between the flow experience and the following sub-aspects of the Qi experience: Feeling of Qi ($M = 3.52$, $SD = 0.75$), Mind ($M = 3.61$, $SD = 0.74$), Qi and Consciousness ($M = 3.55$, $SD = 0.78$), Physical, Mental, and Spiritual Benefits ($M = 3.54$, $SD = 0.77$), and Feeling of Tao ($M = 3.42$, $SD = 0.81$). Moreover, there was a positive correlation coefficient of 0.77 ($p = 0.00$) between the flow experience and the feeling of deep Qi experience (Feeling of Tao).

**Table 5. The relationship between the flow experience and the Qi experience.**

| Aspects | F | AE | CG | CSB | CT | PC | UF | AAM. | TT | LSC | QI | FQ | M | QC | PMSB | deepQi |
|---|---|---|---|---|---|---|---|---|---|---|---|---|---|---|---|---|
| **Flow experience (F)** | 1 | | | | | | | | | | | | | | | |
| Autotelic experience (AE) | .87** | 1 | | | | | | | | | | | | | | |
| Clear Goals (CG) | .82** | .68** | 1 | | | | | | | | | | | | | |
| Challenge-Skill Balance (CSB) | .89** | .75** | .77** | 1 | | | | | | | | | | | | |
| Concentration on Task (CT) | .85** | .74** | .62** | .80** | 1 | | | | | | | | | | | |
| Paradox of Control (PC) | .92** | .77** | .77** | .86** | .81** | 1 | | | | | | | | | | |
| Unambiguous Feedback (UF) | .91** | .75** | .83** | .86** | .76** | .89** | 1 | | | | | | | | | |
| Action–Awareness Merging (AAM) | .90** | .72** | .81** | .83** | .73** | .88** | .88** | 1 | | | | | | | | |
| Transformation of Time (deep flow) (TT) | .78** | .65** | .52** | .61** | .60** | .65** | .63** | .62** | 1 | | | | | | | |
| Loss of Self-Consciousness (deep flow) (LSC) | .88** | .78** | .63** | .70** | .74** | .76** | .71** | .73** | .75** | 1 | | | | | | |
| **Qi experience (QI)** | .90** | .87** | .76** | .82** | .79** | .84** | .83** | .81** | .68** | .78** | 1 | | | | | |
| Feeling of Qi (FQ) | .86** | .83** | .72** | .79** | .77** | .80** | .79** | .76** | .63** | .74** | .93** | 1 | | | | |
| Mind (M) | .83** | .84** | .67** | .74** | .74** | .76** | .73** | .70** | .65** | .74** | .95** | .85** | 1 | | | |
| Qi and Consciousness (QC) | .86** | .82** | .73** | .78** | .74** | .81** | .80** | .78** | .66** | .74** | .95** | .85** | .87** | 1 | | |
| Physical, Mental, and Spiritual Benefits (PMSB) | .87** | .82** | .75** | .79** | .74** | .82** | .82** | .80** | .65** | .75** | .96** | .85** | .88** | .91** | 1 | |
| Feeling of Tao (deep Qi) | .77** | .72** | .69** | .69** | .65** | .71** | .74** | .71** | .57** | .68** | .86** | .73** | .77** | .79** | .80** | 1 |

$N = 58$, $p < 0.01^{**}$.

On the other hand, the seven aspects of the flow experience and the Qi experience were highly correlated, with coefficients ranging from $r = 0.76$ to $0.87$ ($p = 0.00$). These seven aspects are as follows: Autotelic Experience ($M = 3.55$, $SD = 0.76$), Clear Goals ($M = 3.60$, $SD = 0.60$), Challenge–Skill Balance ($M = 3.48$, $SD = 0.62$), Concentration on Task at Hand ($M = 3.48$, $SD = 0.70$), Paradox of Control ($M = 3.57$, $SD = 0.66$), Unambiguous Feedback ($M = 3.48$ $SD = 0.64$), Action–Awareness Merging ($M = 3.54$, $SD = 0.66$). Transformation of Time ($M = 3.25$, $SD = 0.72$) and Loss of Self-Consciousness ($M = 3.45$, $SD = 0.69$) were in general correlated with Qi experience, which coefficients ranging from $r = 0.68$ to $0.78$ ($p = 0.00$). A deep flow (Transformation of Time and Loss of Self-Consciousness) seems to be closely connected to a deep Qi (Feeling of Tao) ($r = 0.57$ to $0.68$, $p = 0.00$), and for more information of data shown in S1 Appendix.

Based on the statistical analysis, the study indicated a high correlation between the flow and Qi experiences, which the conceptualization (Table 1) could be examined by statistical analysis in flow and Qi experiences (Table 5). First, the feeling of Qi is highly related to Autotelic Experience ($r = 0.83$), Paradox of Control ($r = 0.80$), and Challenge–Skill Balance ($r = 0.79$). As one practices Qi well, they feel a sense of the Qi experience. Second, the mind presence in the Qi experience is similar to the Autotelic Experience ($r = 0.84$), Paradox of Control ($r = 0.76$), and Loss of Self-Consciousness ($r = 0.74$), i.e., when one controls the Qi state, one feels an optimal experience from practicing Qi. Third, Qi consciousness is linked to Autotelic Experience ($r = 0.82$), Paradox of Control ($r = 0.81$), Unambiguous Feedback ($r = 0.80$), and Action–Awareness Merging ($r = 0.78$), which concerns controlling one's thoughts and actions without worrying and, while merging body movements with Qi, feeling a sense of Qi and the loss of oneself. Fourth, the physical, mental, and spiritual benefits are similar to the Paradox of Control ($r = 0.82$), Autotelic Experience ($r = 0.82$), and Unambiguous Feedback ($r = 0.82$), which lead to the recovery of oneself and to an experience of the state of flow and the Qi. The last phase in the Qi experience is Tao, which is positive related to Unambiguous Feedback

($r$ = 0.74), Autotelic Experience ($r$ = 0.72), and Action–Awareness Merging ($r$ = 0.71); how-ever, it is also similar to deep flow and Loss of Self-Consciousness ($r$ = 0.68). As one loses one-self and merges with the surroundings without concern for presentation, spiritual feelings, interaction with nature, and even the Tao can be experienced. Other aspects of the flow—for example, Clear Goals, Challenge–Skill Balance, Concentration on Task at Hand, and Transfor-mation of Time—also appear in the state of Qi.

## Discussion

The aim of our study was to examine the relationship between the Qi experience and the flow experience; that is, how ancient East Asian disciplines associated with the Qi experience are related to the Western concept of the flow experience. However, the scope of this study did not originally include a comparison of the discriminating traits and tools of the Qi experience and the flow experience. From the definition in Qi experience and flow experience, we strongly believe the two experience share similar traits. Therefore, we used Table 1 to systematically construct the conceptual validity, and Tables 2–4 to demonstrate reliability and validity. The conceptual validity presented in Table 1 shows a similarity in the definitions, concepts, and traits of the Qi experience and the flow experience. In addition, the study invited both Qigong masters and researchers to construct the content validity using the multi-trait-multi-method (MTMM), verify reliability and validity in the Qi experience (Tables 2 and 3), and re-analyze the consistency of the flow experience (Table 4). The Qi experience and the flow experience both demonstrated a highly acceptable level of reliability that indicated high consistency in their concepts. Moreover, based on Qigong activity, the overall correlations showed a strongly positive relationship between Qi experience and flow experience.

### A possible explanation for the results

The results obtained in the present study yielded much support to our hypothesis that the Qi experience and the flow experience both provide similar inner feelings to those of Qigong practice. We presented the reasons for the high correlative relationship among the aspects of the flow and Qi experiences. When participants immerse themselves in Qigong, they feel a sense of self-growth and satisfaction. Also, in this process, they achieve positive physical and mental outcomes, such as refreshing their bodies and minds, and even experience spiritual thoughts. Few participants, however, obtained the state of deep flow, deep Qi, or the experi-ence of *feeling the Tao*. The definition of *autotelic experience* suggests that the more partici-pants feel happiness or self-growth, the more likely they are to feel tranquility and a sense of Qi, and to want to recapture those feelings. This is consistent with Csikszentmihalyi, who sug-gested that the more skill people have, the more happiness they get from Qigong [8]. More-over, participants may feel a sense of Qi flowing in their body and an enrichment in their mind and spirit. This finding is in line with previous studies [19,31]. The Qi experience may also lead to a peak experience [19]. In this study, we found a connection with Csikszentmiha-lyi's concept that "Qigong exercise concentrates on self- consciousness that entices the Qi to flow" [7]. This is consistent with Enz, who stated that when participants achieve deep flow and deep Qi, they may feel a sense of spirituality and Tao [6].

The findings lead us to believe that there is a positive relationship between deep flow, such as the aspects of *transformation of time* and *loss of self-consciousness*, and the Qi experience (*feeling of Tao*). When a participant loses self-consciousness in a short period of time or feels a difference in the passage of time, Tao and a higher level of happiness from Qigong activity may be realized. The findings support earlier research [2,5,6] indicating that a person who practices spiritual Qigong may feel a loss of self-consciousness and achieve the state of Tao.

Tao is the concept of humanity and follows the laws of nature. This is similar to *TianrenHeyi* (天人合一), which integrates the harmony between humans, nature, and the world [1,6].

## Limitations and recommendations for future research

This study addressed the question of the relationship between flow and Qi when practicing in an urban green environment, albeit with some limitations. The first limitation concerns the assessment used in the current study. Despite having employed a large number of participants and multiple task times, a control group was not used. However, the data nonetheless lend support to idea central to both the Qi experience and the flow experience. The criteria that were used to recruit participants attempt to avoid the placebo effect. Moreover, the results led us believe there were difference between novice (practicing under half a year) and master (practicing more than six years) practitioners in their experience of the sense of Qi, indicating that the assessment process was reliable.

Urban green space plays a critical role in supporting the flow and Qi experiences; however, the differences between outdoor and indoor environments were beyond the main scope of this paper. In addition, Qigong practice locations were different than usual, due to the statement that practicing outdoors is better for "breathing" [1]. According to this statement of Tao and *TianrenHeyi*, the term *biophilia*, coined by Wilson [34], suggests that humans have a deep affiliation to nature rooted in our biology and would rather engage in the natural environment than in an urban environment, which might be another explanation for why Qi participants experience Qi and flow more readily in an urban green environment. By using qualitative methods (see Table 1) to develop the Qi experience questionnaires, we found that participants believed that an outdoor environment with trees, sunlight, and a comfortable microclimate induces a better Qi and flow experience than an indoor environment with no natural elements.

The third limitation is rooted in the daily microclimate in different study sites, which might affect Qigong practice but were not directly controlled in this study. However, the study period was in the early spring to avoid wind, humidity, or hot.

It is challenging to depict our findings, but they could be further examined and widely applied in the future. A suggestion for future studies is to precisely control extraneous variables, such as physical indices that might influence the results. Investigating these physical indices—for instance, heart rate (HR), heart rate variability (HRV), electroencephalography (EEG), and electromyography (EMG)—could be accomplished by having practitioners wear wearable devices while practicing Qigong. Doing so could further help in understanding the relationship between landscape, flow, and the Qi experience. Also, comparing different types of outdoor environments would enhance understanding of how urban green spaces influence the Qi experience and the flow experience.

## Conclusions

The Western concept of the flow experience in outdoor activities (Qigong) seems to be the same as that emphasized in East Asian disciplines, i.e., the Qi experience. This study thus sought to demonstrate the similarities between the feelings evoked by the Qi experience and the flow experience. It was concluded that the Qi experience, as a concept, can be partially explained by the flow experience. This is because while the flow experience follows a measurable scale, Qi is a vague concept, one which nevertheless truly exists in the environment. In this study, the Qi experience was explored via Qi practitioners' perspectives and depictions of their psychological feelings toward interaction with the environment. The study first used qualitative methods to generate the Qi experience questionnaire, after which quantitative

methods were used to test practitioners' Qi experience while practicing, thereby better understanding the Qi experience. The study also yielded some contributions to the development of the Qi experience.

The study converted qualitative answers to quantitative questionnaires in order to more empirically measure the scale of the Qi experience. From the qualitative methodology, the study inferred that the spiritual level of Qi practitioners encompassed feelings of *TianrenHeyi*, happiness, and Tao. From the quantitative methodology, the results showed that when a Qi practitioner goes into a deep flow experience, he/she feels the *transformation of time* and *loss of self-consciousness*. These concepts are linked to a sense of integration with the environment, a sense of transformation of time, a sense of concentration, and a sense of autotelic experience, which are all part of the flow experience. Therefore, these findings could be usefully applied to better understand the concept of the Qi experience and the flow experience.

Moreover, to understand the Qi and flow experiences in an urban green space, it is essential to connect Eastern and Western science in a more holistic and environmentally conscious way for the benefit of humans. The results of this study demonstrate that humans act as a Qi sensor to detect perceptions of the Qi experience and the flow experience in urban green spaces. Future research is required to test how the surrounding environment affects one's psychological state and to contribute more evidence connecting Western and Eastern disciplines.

## Supporting information

**S1 Appendix. This is the Qi experience and flow experience file.**
(XLSX)

## Acknowledgments

This journal article is modified parts from the master's thesis of *Effect of Landscape Types on Flow experience and Qi Experience*, Department of Horticulture and Landscape, National Taiwan University. The authors thank Editor Dr. Michael B. Steinborn and reviewers Dr. Roger Jahnke, Dr. M. Sc. Jonas Ebert, Dr. Sonja Annerer-Walcher, and one anonymous reviewer for their valuable and insightful feedback and their time and effort on an early draft of this manuscript.

## Author Contributions

**Conceptualization:** Chun-Yen Chang.

**Data curation:** Shih-Han Hung.

**Methodology:** Shih-Han Hung.

**Resources:** Ching-Yung Hwang.

**Supervision:** Ching-Yung Hwang, Chun-Yen Chang.

**Writing – original draft:** Shih-Han Hung.

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
