## [Decision Letter · Decision Letter 0]

15 Jun 2020

PONE-D-20-11330

Is the Qi Experience related to the Flow experience? Practicing qigong in urban green spaces

PLOS ONE

Dear Dr. Chang,

Thank you for submitting your manuscript to PLOS ONE. Four experts commented on your manuscript. As you can see from the reviews, all referees found the general topic addressed in your manuscript interesting. At the same time, all reviewers found that there is need for more explanation and systematic evaluation regarding aspects of construct validity and measurement methodology. While this will call for some more efforts, I consider it worthwhile. Taken together, I would like to invite you preparing a revision that addresses the remaining concerns, together with a cover letter that provides point-to-point replies. Some more detailed editor's comments are provided below. 

Best regards,Michael B. Steinborn, PhDAcademic Editor

We look forward to receiving your revised manuscript.

Kind regards,

Michael B. Steinborn, PhD

Academic Editor

PLOS ONE

Journal Requirements:

2. We note that Figures 1 a-c include an image of participants in the study. 

As per the PLOS ONE policy (http://journals.plos.org/plosone/s/submission-guidelines#loc-human-subjects-research) on papers that include identifying, or potentially identifying, information, the individual(s) or parent(s)/guardian(s) must be informed of the terms of the PLOS open-access (CC-BY) license and provide specific permission for publication of these details under the terms of this license.

Please download the Consent Form for Publication in a PLOS Journal (http://journals.plos.org/plosone/s/file?id=8ce6/plos-consent-form-english.pdf). The signed consent form should not be submitted with the manuscript, but should be securely filed in the individual's case notes.

Please amend the methods section and ethics statement of the manuscript to explicitly state that the patient/participant has provided consent for publication: “The individual in this manuscript has given written informed consent (as outlined in PLOS consent form) to publish these case details”.

If you are unable to obtain consent from the subjects of the photographs, you will need to remove the figures and any other textual identifying information or case descriptions for these individuals.

Editor's comments 

 (--1--) Methodology of Evaluating Tests and Questionnaires In general, all four reviewers pointed to the need of a more systematic evaluation of construct validity regarding the concepts of Qi and Flow. I would recommend presenting data and results in a more systematic fashion, using the multi-trait-multi-method (MTMM) matrix, which is a traditional approach to assessing the construct validity of a set of variables/measures related to theoretical concepts. Campbell and Fiske (1959) provided a practical methodology to be used for purposes of construct validation and the development of tests and questionnaires. At its core are two types of validity termed convergent and discriminant, which can be viewed as subcategories of construct validity. Convergent validity refers to the degree to which concepts that should be related in theory are actually interrelated through empirical observation. Discriminant validity refers to the degree to which concepts that not related in theory are actually not interrelated empirically. In order to determine construct validity of your target measure, you would have to determine both convergence and discrimination. Key to the evaluation of construct validity, however, is the analysis of test reliability as reliability coefficients are a baseline against which to interpret cross-correlations. In some way, reliability (in terms of accuracy of measurement) can be interpreted as a prerequisite to validly measure relationships with other constructs.  (--2--) Evaluation of the concept of Qi As R2 indicates, it is important to show at the same time what Qi is and what it is not. More practically, in order to defend a proposition about what a test measures (in your case Qi), one looks basically for two things. The first is convergence of indicators. Let me illustrate this point. For example, in order to justify that a questionnaire measures Qi, the concepts is first defined analytically (through thinking about what it is) and then is set into a hypothetical space of related concepts, like for example, flow state, relaxation, or other related hypothetical states. In the next step, one has to collect data from other instruments claimed to measure these closely related constructs and to compare them by means of correlation analysis. The second point refers to discrimination. That is, scores identified with supposedly distinct states should not correlate to highly. For example, a test claimed to measure "creative state" or "alertness" should reveal some but not too high correlations with the target construct (Qi) and is therefore interpreted as to indicate discriminant validity. Another point refers to aspects of measurement accuracy. Crucially, the MTMM logic (see Campbell & Fiske, 1959) requires that measures are basically reliable, which again is dependent on the number of measurement units under the score (see Steinborn, Langner, Flehmig, & Huestegge, 2018). Low or even zero correlations cannot be interpreted as indicating discriminant validity if one or both measures lack test reliability. In other words, convergent and discriminant validity can only be interpreted by simultaneously considering reliability as the underlying basis. This means that low correlations can indicate both independence of the constructs and/or low reliability of (at least one) of the constructs. With regard to your study, I would suggests reworking the manuscript towards a more systematic step-by-step evaluation following the standard methodology of concept validation. [suggested literature: Campbell, D. T., & Fiske, D. W. (1959). Convergent and discriminant validation by the multitrait-multimethod matrix.  Psychological Bulletin, 56(2), 81-105. doi:10.1037/h0046016; Steinborn, M. B., Langner, R., Flehmig, H. C., & Huestegge, L. (2018). Methodology of performance scoring in the d2  sustained-attention test: Cumulative-reliability functions and practical guidelines. Psychological Assessment,  30(3), 339-357. doi:10.1037/pas0000482].  (--3--) Theory of Qi An essential point concerns the distinction of the concept of Qi from a theoretical perspective. All reviewers pointed out that the concepts of Qi seems important but at the same time is in need of a more systematic theoretical evaluation. What is it that is meant by Qi relative to Flow? Also, it should be evaluated whether the correlation between Qi and Flow is due to the similarity of items under the scores of both concepts (this point was raised by R3 and R4), and if so, whether this is intended, not intended, or due to a natural conceptual overlap between the concepts. I suggest elaborating more on the theoretical nature of the concepts of Qi and Flow and delineating ways of how to measure both concepts empirically. 

Reviewers' comments:

Reviewer's Responses to Questions

**Comments to the Author**

1. Is the manuscript technically sound, and do the data support the conclusions?

Reviewer #1: Yes

Reviewer #2: Partly

Reviewer #3: Partly

Reviewer #4: Partly

2. Has the statistical analysis been performed appropriately and rigorously? 

Reviewer #1: Yes

Reviewer #2: No

Reviewer #3: I Don't Know

Reviewer #4: No

3. Have the authors made all data underlying the findings in their manuscript fully available?

Reviewer #1: Yes

Reviewer #2: No

Reviewer #3: No

Reviewer #4: Yes

4. Is the manuscript presented in an intelligible fashion and written in standard English?

Reviewer #1: Yes

Reviewer #2: No

Reviewer #3: No

Reviewer #4: Yes

5. Review Comments to the Author

Reviewer #1: There are a few edits in language that should be done.

I appreciate this article because, usually the concept of Qi -- important but not understood in the West -- is discounted. This article attempts to bring Qi into the dialogue.

This is an excellent contribution.

Reviewer #2: Background:

The study aims to examine if the western concept of Flow is related to the eastern concept of Qi. To this end, the authors created a Qi-questionnaire. Subjects had to fill-out this questionnaire after doing Qi-Gong as well as a modified Flow-Questionnaire. The correlation between the two questionnaires shows a very high correlation showing a conceptual overlap of the two concepts.

I enjoyed reading the document and was especially intrigued by the idea of similar psychological mechanisms in western and eastern concepts. Furthermore, I applaud the effort to expand the psychological toolbox by constructing new questionnaires. However, I would recommend improving the document in several aspects including literature analysis, structure, conceptual analysis, methodological description, and discussion. I can recommend this work after major revisions. Please see my comments below for further detail.

[1] Conceptual analysis

In my opinion, the paper lacked a deeper analysis of the two concepts in question. When comparing psychological mechanisms a deep analysis and precise definitions are helpful for the reader better to understand the comparison. This is especially important when dealing with (to western audiences) unfamiliar concepts like Qi, Tao, and Qigong. For example, I was wondering about what physical activity is Qigong, especially in comparison to Thai Chi, Yoga, or similar activities.

[2] Literature analysis

In addition to [1], extended literature analysis could help to clarify definitions and concepts. Especially for well-known concepts like Flow, I would expect a large amount of helpful scientific literature. I could imagine that a similar relationship like the one focused on in this paper can be found regarding other physical activities as well. It would be interesting, for example, if western sports evoke Flow or to analyze the positive benefits of Qigong, Yoga, Tai Chi, etc. Overall, I would recommend expanding the literature.

[3] Structure

The reading, although a good read overall, was sometimes difficult to follow. I would recommend improving the structure of arguments for example by introducing more headlines and straighten the line of arguments. Sometimes different paragraphs would discuss the same topic or paragraphs would switch their line of argument unexpectedly to different content.

[4] Table 1

I stumbled about table 1’s content and especially its size. It seems the content of table 1 can on the one hand be discussed in the continuous text (comparison Qi – Flow) or on the other hand, can be presented in the annex (open-ended data). Perhaps the 14 questions regarding the different aspects of Qi are the most suitable content for table 1.

[5] Methodological concerns

I wonder how exactly the data collection as well as the correlation analysis between Qi and Flow was done, as the obtained correlations are incredibly high. Unfortunately, the description of the data collection and the statistical analysis is rather short. I suggest providing some more information with this regard. The following methodological issues should be discussed or at least reported:

Sequencing effects could influence the subjects’ judgments. It is however unclear in what order subjects answered the two questionnaires.

Expectation effects could influence judgments. However, it is unclear how the instructions were presented or how the subjects were informed about the aim of the study.

Selection effects could also be a problem. However, it is not reported on what basis subjects were selected or even how subjects were recruited.

Translation effects could be a huge problem. Both questionnaires used the word "Qi" and are therefore likely to measure the same concept (at least to some degree).

The validity of data, exclusion criteria, and missing data could play a role but are not reported.

The main finding is the very high correlation between Qi and Flow experience but it is not clear how these scores were calculated from the sub scores.

[6] Conclusion

The conclusion could be expanded to incorporate different explanations for the findings of the paper. In correlation designs, several types of causal (or not causal) links are possible. The structural relationship could be:

a) Flow is a part of the Qi-Gong experience (among others).

b) Flow and Qi are the same concept.

c) Qi-Gong evokes Flow depending on the skill of the practitioner.

d) Flow evokes Qi.

e) Physical activity (of any kind) leads to both Flow and Qi.

Probably several other connections are also plausible. These should be explored and discussed. Additionally, future research directions to address these explanations could prove interesting.

[7] Grammar and formulations

While reading the paper I regularly stumbled upon punctuation mistakes or unfamiliar grammatical structures and sentences. Having studied the Chinese language myself, I know about the huge differences in grammar and wording compared to western languages. I would recommend the help of a native speaker to reduce linguistic mistakes.

For example line 62 “In Taoists“ should be “In Taoist philosophy”; “In Taoism” or “For Taoists”.

[8] line comments

45 “the mental state that influences overall health”: there is not just one mental state that can influence overall health.

58 “sense of tao”: this concept should be explained for readers unfamiliar with Chinese philosophy

70 “represent special forms of the Flow experience”: Are there studies that analyzed this? If yes, you can go into detail since this seems very important for your research question. If no, on what is this claim based? And what does “a special form of the Flow experience” mean?

76 “the urban green environment”: is that some form of organization?

126 “the words “Qi experience,” “Qi,” and “Qi practice” were added”: This could be a big problem. If you change the wording from “Flow” to “Qi” you probably don’t measure Flow anymore but Qi instead. Perhaps you should repeat the analysis without these items.

142 “were selected”: How did the selection process work? What were the selection criteria?

146 ”letter of Qi and environmental experience consent”: Please explain what this consent includes

148 “from 7:00–11:00”: It does not seem necessary to go into detail about the theoretical foundation of the timeslot.

153 “After practicing, breathing for 10 minutes, finishing the questionnaires”: The description of the procedure is quite short and does not explain the procedure completely understandable. Please expand this point in a separate paragraph. Did the participants do Qigong for 4 hours or just at a certain time between 7 and 11 a.m.? What amount was the gift card and what for? In what order were the questionnaires answered? etc.

161 “F=3.32”: What kind of analysis was used?

188 “Furthermore, an outdoor environment with trees … induces a better Qi and Flow”: It is unclear where this insight comes from. Is it a quote, a result, or an intuition?

192 “Since urban green space plays a critical role in”: Urban green spaces are mentioned regularly but it is not clear why they are important to the research question. Qi as well as Flow can be experienced indoors.

Table 3 “N=654”: I am confused about the sample size. You wrote about 58 subjects. Therefore, N should be 58.

235 “used mixed methods”: It is unclear what mixed methods where used. The main body of the research seems to be based on questionnaires only.

Bibliography: Not in alphabetical order

Reviewer #3: The submitted study investigated the relationship between the eastern concept of Qi experience and the western concept of flow during Qigong. In a first step, they used qualitative methods to generate a questionnaire that assesses the Qi experience. In a second step, they assessed the Qi experience and the flow experience after Qigong practice in a group of students and Qigong masters in three different environments. They found high correlations between scales of the Qi experience questionnaire and the scales of the Flow State Scale, suggesting a large overlay of the concepts of Qi experience and flow during Qigong.

The submitted study would enrich the present literature on Qi experience and flow. However, the manuscript would need some major revisions before being ready for publication. Especially, the manuscript would benefit from a clearer structure and more essential details in the methods and results section. This would also allow me to better evaluate the quality of the methods and analysis.

1. I would welcome a clearer structure in the introduction and discussion.

2. As you have “urban green spaces” in your title, please introduce their role for the Qi experience in the introduction.

3. Please add details and structure to the methods section.

4. For example, it is not clear how many students were asked the open questions for the generation of the Qi experience questionnaire. Were there two groups of 26 students each, so 52 in total? Were they asked once only, once after practice in each of the three environments, or even multiple times in each environment?

5. Participants in the questionnaire study: Please add some details to this section. Did participants participate for only one session or multiple times? Were they always assigned to the same site or did the site change for participants? How often did they participate on average? Were students and masters evenly distributed across sites? Were the students the same as in the open-ended questions study?

6. Results: “A total of 654 valid data were collected (n=654 per person-time)”. Does this mean there were 654 completed questionnaires packages overall?

7. In the last sentence of the results section you mention that the outdoor environment had a positive effect. Please add data and results of this analysis.

8. Although you state that all data is available, I could not find any reference that would allow me to access the data.

Reviewer #4: Qi experience is an interesting topic. The study analyszed the relationship between the Qi experience and the flow experience by using quantitative and qualitative methods. However, the result of correlation between the flow and Qi experiences is insufficient to support the final conclusion, especially for the placebo effect.

Method:

1. Data were collected 12 times within the last spring and late fall, and in different practice environments, including (a)indoors, (b) a built environment with nature, and (c) the natural environment at NTU (line 85-87). According to the authors’ description, Qigong is a type of mind-body exercise that emphasizes a harmonious interaction between humans and the environment (line 54-55).

Are there any differences of the data among different environment or among different time points?

2. The Flow State Scale was translated into Chinese (line 126). Has the Chinese version been revised and tested for reliability and validity?

Results:

3. Please check which p value is correct: For the number of years practicing Qigong, the results show significant 161 influence on the flow experience (F=3.32* p<0.05, p <0.01, η2=0.010) (line 161).

4. The p values should not be 0.0. (line 162). Maybe you can use p<0.001 instead.

5. There is an important problem that all the subjects have received Qigong training, and the description of Qi is also included in The Flow State Scale. Without the control group, the subjects may know the purpose of the study, and be affected by the placebo effect.

6. PLOS authors have the option to publish the peer review history of their article (what does this mean?). If published, this will include your full peer review and any attached files.

Reviewer #1: Yes: Roger Jahnke, OMD

Reviewer #2: Yes: Jonas Ebert (M.Sc.)

Reviewer #3: Yes: Sonja Annerer-Walcher

Reviewer #4: No

---

## [Author Response · Author response to Decision Letter 0]

6 Aug 2020

Thank you for inviting us to submit a revised draft of our manuscript entitled, “[Is the Qi Experience Related to the Flow Experience? Practicing Qigong in Urban Green Spaces],” to [PLOS ONE]. We really appreciate the time and effort you and each of the reviewers have dedicated to providing insightful feedback on ways to strengthen our paper. 

It is with great pleasure that we resubmit our article for further consideration. We have included a point-by-point response to the questions and comments delivered in your letter dated June 16, 2020 in the cover letter. The revised words and sentences are highlighted in gray in the track file. 

We hope that the revised manuscript is clearer in its conceptual analysis of the Qi and flow experiences. We hope these revisions satisfactorily address all the issues and concerns you and the reviewers have noted. Again, thank you.

---

## [Decision Letter · Decision Letter 1]

22 Sep 2020

Is the Qi Experience related to the Flow experience? Practicing qigong in urban green spaces

PONE-D-20-11330R1

Dear Dr. Chang,

We’re pleased to inform you that your manuscript has been judged scientifically suitable for publication and will be formally accepted for publication once it meets all outstanding technical requirements.

Kind regards,

Michael B. Steinborn, PhD

Academic Editor

PLOS ONE

Additional Editor Comments (optional):

Reviewers' comments:

Reviewer's Responses to Questions

**Comments to the Author**

1. If the authors have adequately addressed your comments raised in a previous round of review and you feel that this manuscript is now acceptable for publication, you may indicate that here to bypass the “Comments to the Author” section, enter your conflict of interest statement in the “Confidential to Editor” section, and submit your "Accept" recommendation.

Reviewer #3: (No Response)

Reviewer #4: All comments have been addressed

2. Is the manuscript technically sound, and do the data support the conclusions?

Reviewer #3: Yes

Reviewer #4: Partly

3. Has the statistical analysis been performed appropriately and rigorously? 

Reviewer #3: Yes

Reviewer #4: Yes

4. Have the authors made all data underlying the findings in their manuscript fully available?

Reviewer #3: Yes

Reviewer #4: Yes

5. Is the manuscript presented in an intelligible fashion and written in standard English?

Reviewer #3: No

Reviewer #4: Yes

6. Review Comments to the Author

Reviewer #3: The authors thoroughly revised the manuscript and enriched the manuscript with more details and data.

1. However, the readability of the manuscript should be further improved to make this valuable work more accessible for the audience. I have troubles understanding some of the added paragraphs. Maybe some words were mixed up or rephrasing of those sentences makes them easier to read. Further, splitting up long sentences can improve readability. For example, line 103 – 105, or the long sentence line 137 – 143. I myself find it hard to write down my work in nicely readable sentences, as I am a non-native English speaker. Maybe you can find someone who checks your text and helps you increase readability? It would make your valuable work more accessible for a broader audience.

2. In line 419 – 422 you write “By using quantitative methods to develop the Qi experience questionnaires, we found that participants believed that an outdoor environment with trees, sunlight, and a comfortable microclimate induces a better Qi and flow experience than an indoor environment with no natural elements.” However, I could not find any quantitative data supporting this sentence in your manuscript. Maybe you meant “qualitative” data in table 1?

3. In line 423 you write that the different study sites are a limitation of your study. I think the many different study sites are a strength of your study. Because, by having many sites, the general effect of Qi-Gong practice is less masked by site-specific effects.

4. Line 75: you wrote „Dao“, shouldn’t it mean “Tao”?

Reviewer #4: (No Response)

7. PLOS authors have the option to publish the peer review history of their article (what does this mean?). If published, this will include your full peer review and any attached files.

Reviewer #3: **Yes: **Sonja Annerer-Walcher

Reviewer #4: No

---

## [Editor Report · Acceptance letter]

7 Oct 2020

PONE-D-20-11330R1 

Is the Qi Experience related to the Flow Experience? Practicing qigong in urban green spaces 

Dear Dr. Chang:

I'm pleased to inform you that your manuscript has been deemed suitable for publication in PLOS ONE. Congratulations! Your manuscript is now with our production department. 

Kind regards, 

on behalf of

Dr. Michael B. Steinborn 

Academic Editor

PLOS ONE